# An Improved Dual Second-Order Generalized Integrator Phased-Locked Loop Strategy for an Inverter of Flexible High-Voltage Direct Current Transmission Systems under Nonideal Grid Conditions

**Lai Peng [1], Zhichao Fu [1], Tao Xiao [1], Yang Qian [2,*], Wei Zhao [3] and Cheng Zhang [1]**

[1] Guangdong Power Grid Co., Ltd., Guangzhou Power Supply Bureau, Guangzhou 510630, China; lf19971003@foxmail.com (L.P.)
[2] Nanjing Institute of Technology, Nanjing 211167, China
[3] NR Electric Co., Ltd., Nanjing 211102, China
* Correspondence: y00450200520@njit.edu.cn

**Abstract:** High-voltage flexible power systems, with their intrinsic characteristics, play an increasingly important role in electronic power systems. Synchronization between the inverter and the grid needs to be achieved by a phase-locked loop (PLL), the performance of which determines the quality of power transmission. This paper proposes a PLL adapted to extremely harsh grid conditions. Firstly, the traditional synchronous reference frame PLL and the dual second-order generalized integrator (DSOGI-PLL) are analyzed, and the errors in phase-locking and the shortcomings of these two methods in the presence of DC components in the grid are pointed out. Secondly, based on the harmonic grid voltage, a repetitive control internal model is introduced by DSOGI to realize the real-time tracking and regulation of the harmonic signals in order to suppress the harmonic voltage disturbance. In addition, a DC bias elimination and frequency adaptive method is proposed to solve the problems of DC bias and grid voltage frequency fluctuation in order to achieve adaptive tracking of the grid phase. Finally, the superiority of the proposed strategy is verified through simulations and experiments.

**Keywords:** flexible DC transmission; PLL; unbalance voltage; harmonic voltage; DC bias; power quality

## 1. Introduction

Energy sources serve as the foundation for economic development. However, the reserves of traditional energy resources, such as coal and petroleum, are limited, and their consumption can lead to environmental pollution. As a result, there has been a growing focus on the development of clean and renewable energy sources, both domestically and abroad [1,2]. Clean energy sources are often located far away from their load centers and the main grid, with limited installed capacity due to environmental constraints. Therefore, high-voltage direct current (HVDC) transmission is considered to have significant economic, technological, and other advantages over traditional AC transmission [3].

The voltage source converter (VSC) control on the grid-connected side is composed of two primary branches: the synchronization scheme and the inner-loop current control. The synchronization scheme requires accurate phase locking to the grid voltage and serves as a prerequisite for control. The most commonly used device for synchronization is the synchronous reference frame-based phase-locked loop (SRF-PLL) [4]. Known for its simple structure, the SRF-PLL is capable of accurately locking the phase in an ideal grid. However, in practice, grids often exhibit nonideal characteristics such as imbalances, harmonic waves, DC offsets, and frequency fluctuations [5]. In such cases, the SRF-PLL may be affected by grid voltage during the phase angle acquisition process, resulting in a significant phase

offset error and reducing the control system's performance. Therefore, accurately tracing voltage phase information in a nonideal grid is of utmost importance for engineers.

Many scholars have researched phase-locking methods for use in imbalanced grids. One such method is the decoupled double synchronous reference frame (DDSRF-PLL), which was proposed by a scholar in [6] to separate the positive and negative sequences of voltage signals. However, the DDSRF-PLL method adopts double dq and involves many filters and complicated calculations, resulting in a long transient response time. An alternative approach is the dual second-order general integrator (DSOGI-PLL), which was adopted in [7]. The DSOGI-PLL approach introduces an orthogonal signal generator (QSG) and positive- and negative-sequence component calculation (PNSC) modules based on SRF-PLL. The positive-sequence component of the grid voltage is extracted as the input signal of SRF-PLL for phase angle locking, leading to good results. Scholars proposed an instantaneous sequence component extraction scheme in [8] to accelerate the phase-locking speed. In contrast, the instantaneous sequence component extraction scheme does not require the closed-loop detection parameter setting and calculates the instantaneous voltage sequence component directly from the grid voltage and virtual orthogonal signal, demonstrating high calculation accuracy. As for traditional methods for phase locking in a grid with harmonic distortion, engineers have improved the design of PLL loop filters and phase discriminators. In the first method, a filtering step is introduced, such as the wave trap [9] and moving average filter [10]. However, its harmonic voltage suppression effect is not ideal.

In the second method, the multiple second-order general integrator [11] (MSOGI-PLL) and the delay signal elimination method [12] are used to extract the positive-sequence voltage of the fundamental wave by constructing an orthogonal signal. This method can effectively reduce the influence of harmonic voltage. However, MSOGI-PLL has a complex structure and algorithm, and the delay signal elimination algorithm has errors and poor robustness. In [13], an internal model for repetitive control was introduced into the cross-decoupled complex filter PLL, which can eliminate harmonic waves completely during the phase-locking process. Each of the above methods has its advantages and disadvantages. To broaden the application scope of PLL, further research is necessary for a phase-locked scheme that can accurately capture phase information under different grid conditions. Some comparative studies of PLL are included in [14].

The phase-locked schemes under various nonideal power grid conditions were researched herein, such as imbalance, harmonic wave, DC offset, and frequency fluctuation; thus, an improved DSOGI-PLL strategy was proposed. In Section 3, the internal model for repetitive control was embedded on the basis of the original PLL. The fundamental wave and the harmonic component were extracted by virtue of the resonance peak at the fundamental wave and each harmonic frequency. Simultaneously, the wave trap was introduced by us to filter the fundamental wave; the harmonic signal was negatively fed back to the input terminal; thus, the harmonic voltage across the frequency band was eliminated. In Section 4, the low-pass filtering link where the voltage error signal was input was introduced into SOGI-QSG so that the DC component of the orthogonal signal could be offset. Finally, the fixed resonant frequency originally set by SOGI-QSG was replaced by the grid angular frequency obtained from the SRF-PLL link. In this way, the adaptive frequency adjustment was realized. In Section 4, through simulation, the accuracy of the proposed synchronization scheme was verified.

## 2. Traditional DSOGI-PLL Scheme

### 2.1. Analysis Error of SRF-PLL

The scope of application of PLL in HVDC systems is as follows. Figure 1 shows the structure of the SRF-PLL, where the phase discriminator is enclosed in a dotted box, and $1/s$ represents the voltage-controlled oscillator. The collected grid voltage information was transformed into the dq coordinate system using the $u_{\mathrm{gq}}$ transmission PI regulator.

The resulting output was then compared with the given frequency value, and the phase information $\hat{\theta}$ was obtained by integrating the difference.

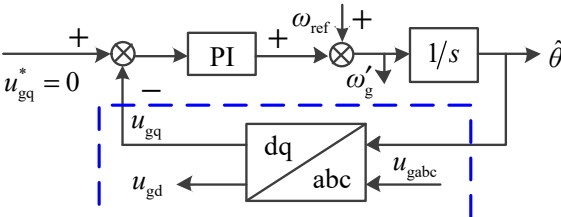

**Figure 1.** Structural diagram of SRF-PLL.

The simple SRF-PLL algorithm is capable of accurately locking the phase information in an ideal power grid. However, in practical scenarios, imbalanced voltages and harmonic waves often occur due to nonlinear loads, asymmetric faults, and other issues, resulting in a nonideal state. In the dq coordinate system, the fundamental wave voltage is represented as a DC value, while the odd harmonic voltage is represented as an AC value, leading to AC interference in the obtained $u_{gq}$. The mathematical model for imbalanced and harmonic grid voltage can be expressed as follows:

$$\begin{bmatrix} u_{ga} \\ u_{gb} \\ u_{gc} \end{bmatrix} = u_g \begin{bmatrix} \cos\theta \\ \cos(\theta - \frac{2\pi}{3}) \\ \cos(\theta + \frac{2\pi}{3}) \end{bmatrix} + u_g \begin{bmatrix} \cos(\theta) \\ \cos(\theta + \frac{2\pi}{3}) \\ \cos(\theta - \frac{2\pi}{3}) \end{bmatrix} + \sum_{\substack{-\infty \\ n \neq \pm 1}}^{+\infty} a u_g \begin{bmatrix} \cos(n\theta) \\ \cos(n\theta - \frac{2\pi}{3}) \\ \cos(n\theta + \frac{2\pi}{3}) \end{bmatrix} \quad (1)$$

where $n$ is odd, and three components represent the voltage of the positive and negative fundamental wave sequences and each odd harmonic wave, respectively. Here, $u_g$ is the amplitude of the fundamental wave voltage, $a$ is the voltage amplitude coefficient, and $\theta$ is the phase angle of the positive-sequence voltage of the fundamental wave.

After applying coordinate transformation, we obtained the expression for $u_{gq}$ as follows:

$$u_{gq} = u_g \sin(\theta - \hat{\theta}) - u_g \sin(\theta + \hat{\theta}) + \sum_{\substack{-\infty \\ n \neq \pm 1}}^{+\infty} a u_g \sin(n\theta - \hat{\theta}) \quad (2)$$

where $\hat{\theta}$ represents the phase angle detected by PLL; there may exist a deviation between the actual values of $\hat{\theta}$ and $\theta$.

If we assume a deviation of 0, (2) can be transformed into

$$u_{gq} = -u_g \sin(2\hat{\theta}) + \sum_{\substack{-\infty \\ n \neq \pm 1}}^{+\infty} a u_g \sin[(n-1)\hat{\theta}] \quad (3)$$

### 2.2. Tradition DSOGI-PLL

The accuracy of the detected phase information depends on $u_{gq}$ being equal to 0. However, based on the above equation, the negative-sequence voltage of the fundamental wave introduces a fluctuation component at twice the frequency of $u_{gq}$, and the $n$-th harmonic voltage introduces a fluctuation component at $(n-1)$ times the frequency of $u_{gq}$, resulting in an error in the locked phase angle.

According to the principle of phase locking, obtaining the positive-sequence component of voltage in the αβ coordinate system (using $u_g$) is a prerequisite for accurate

phase locking [15,16]. Using the symmetric component method, we obtained the following equation:

$$\begin{cases} u_{g\alpha\beta+} = [T_{\alpha\beta}]u_{gabc+} = [T_{\alpha\beta}][T_+]u_{gabc} = \frac{1}{2}\begin{bmatrix} 1 & -q \\ q & 1 \end{bmatrix}u_{g\alpha\beta} \\ u_{g\alpha\beta-} = [T_{\alpha\beta}]u_{gabc-} = [T_{\alpha\beta}][T_-]u_{gabc} = \frac{1}{2}\begin{bmatrix} 1 & q \\ -q & 1 \end{bmatrix}u_{g\alpha\beta} \end{cases} \quad (4)$$

where $[T_+]$, $[T_-]$, and $[T_{\alpha\beta}]$ comprise the Clark transformation matrix; $q = e^{-jI/2}$ indicates a phase shift at 90°.

According to (4), the positive-sequence component of voltage under the $\alpha\beta$ axis can be expressed as

$$\begin{cases} u_{g\alpha+} = \frac{1}{2}(u_{g\alpha} - qu_{g\beta}) \\ u_{g\beta+} = \frac{1}{2}(qu_{g\alpha} + u_{g\beta}) \end{cases} \quad (5)$$

According to the above equation, a phase shift at 90° for $u_{g\alpha}$ and $u_{g\beta}$ should be made. Such a function can be realized by the second-order generalized integrator [17]. Its schematic diagram is displayed in Figure 2.

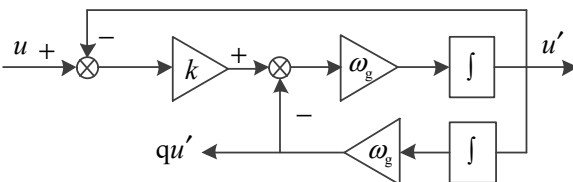

**Figure 2.** The functional block diagram of SOGI-QSG.

Based on the above equation, a 90° phase shift for $u_{g\alpha}$ and $u_{g\beta}$ is required. This function can be achieved using the second-order generalized integrator [17], which is depicted in Figure 2.

The SOGI-QSG transfer function is shown as below:

$$\begin{cases} D(s) = \frac{u'(s)}{u(s)} = \frac{k\omega_g s}{s^2 + k\omega_g s + \omega_g^2} \\ Q(s) = \frac{qu'(s)}{u(s)} = \frac{k\omega_g^2}{s^2 + k\omega_g s + \omega_g^2} \end{cases} \quad (6)$$

where $\omega_g$ is the resonant angular frequency and $k$ is the gain coefficient.

Only when the angular frequency of the signal input in the quadrature signal generator (QSG) link is equal to the resonant angular frequency (i.e., $\omega = \omega_g$) and both $D(s)$ and $Q(s)$ are equal to $1\angle 90°$ can orthogonal signals of equal amplitude be generated. Figure 3a and 3b, respectively, show the Bode diagrams of $D(s)$ and $Q(s)$ under different $k$ values. $D(s)$ represents a band-pass filter, whereas $Q(s)$ represents a low-pass filter. Observing the Bode diagrams, we can notice that the amplitude gains of both under k values are 0 at the resonant frequency point. Thus, there is no offset generated in inputting the fundamental wave component.

Based on Figure 3a, we can observe that the bandwidth of the $D(s)$ filter is affected by $k$. As $k$ decreases, the bandwidth narrows, and the attenuation of signals at other frequencies increases.

From the amplitude–frequency characteristic curve in Figure 3b, we can see that at the resonant frequency of 50 Hz, the gain attenuates at a rate of −40 dB/dec. However, the attenuation of the $Q(s)$ filter to low-frequency signals is very weak. When the frequency is less than 50 Hz, the attenuation decreases with a higher $k$ value.

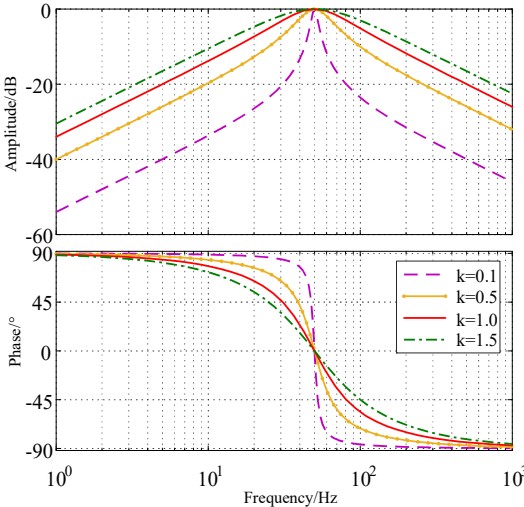

**(a)** Corresponding to *D*(*s*) Bode diagram.

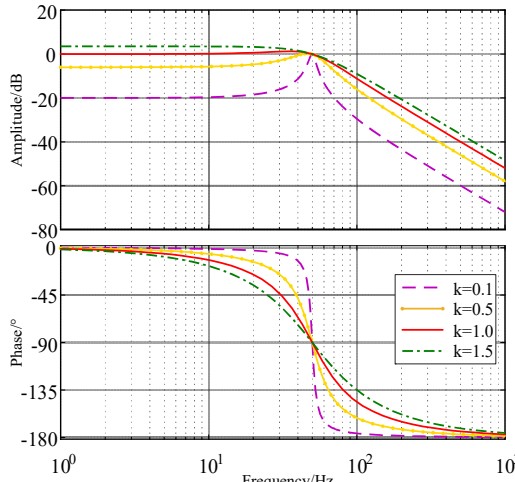

**(b)** Corresponding to *Q*(*s*) Bode diagram.

**Figure 3.** Analysis of frequency spectrum of SOGI-QSG.

It is important to note that both *D*(*s*) and *Q*(*s*) have weak attenuation capacity to low-frequency harmonic waves. When a power grid contains many low-frequency harmonic components, the DSOGI method may not be able to filter out all of these harmonic waves, which may lead to harmonic interference in the voltage signal extraction process and result in phase-locked errors. In addition, although the *D*(*s*) filter can filter out the DC component to some extent, the *Q*(*s*) filter's low-pass characteristic means that it has very weak attenuation performance to DC, causing the DC component to appear in the output signal. The output DC signal is related to the value of *k*. When *k* < 1, the DC output will be attenuated; when *k* = 1, the DC signal will be output without loss; when *k* > 1, the DC component increases. Therefore, the DSOGI-PLL method may not be able to accurately lock the phase information in the event of a DC offset within the grid.

The structural diagram of the traditional DSOGI-PLL is shown in Figure 4. First, the grid voltages, $u_{g\alpha}$ and $u_{g\beta}$, are input into SOGI-QSG, which generates the alternating- and direct-axis voltage signals, $u'_{g\alpha}$, $u'_{g\beta}$, q$u'_{g\alpha}$, and q$u'_{g\beta}$. Then, the voltage sequence components are calculated via the PNSC module. The positive-sequence components, $u^+_{g\alpha}$ and $u^+_{g\beta}$, are input into SRF-PLL to obtain the phase angle $\hat{\theta}$.

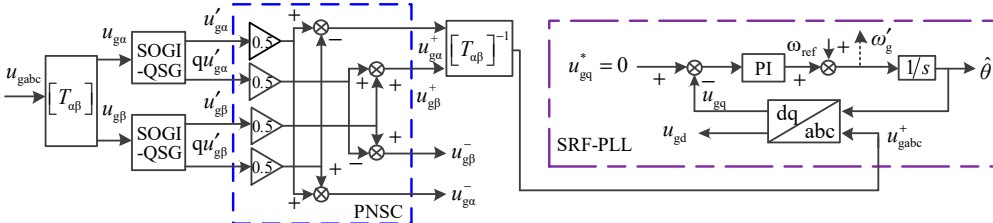

**Figure 4.** Structural diagram of DSOGI-PLL.

## 3. The Improved DSOGI-PLL Scheme

### 3.1. Elimination Strategy Based on Internal Model Control

To address the issue of harmonic voltage interference and improve the accuracy of PLL in a harmonic grid, an internal model for repetitive control was introduced into the traditional DSOGI-PLL outer loop. Based on the principle of repetitive control, internal model control can generate resonant gain at the frequencies of both the fundamental wave and harmonic signals, enabling real-time tracking and regulation of each harmonic signal. This approach allows for periodic regulation and ensures zero-error tracking.

Mathematically, this process can be expressed by the following expression:

$$G(s) = \frac{1}{1 - e^{-Ts}} \tag{7}$$

where $T$ is the period of the external signal. The z transformation was conducted upon (7). According to the transformation principle $z = e^{-Ts}$, the expression in the discrete domain was obtained as follows:

$$G(z) = \frac{1}{1 - z^{-A}} \tag{8}$$

where $A = f_s/f$, $f$ is the fundamental signal frequency and $f_s$ is the sampling frequency.

In a harmonic grid, the odd harmonics such as 5, 7, and 11 have the highest voltage content and can cause the most interference of the PLL. Therefore, efforts should be focused on suppressing only the odd harmonic voltages. An improved internal model structure for the odd harmonic generator was proposed in [13] to enhance the system's response performance. This approach can reduce the delay cycle by half, and the mathematical expression is as follows:

$$G(z) = -\frac{1}{1 + C(z)z^{-A/2}} \tag{9}$$

where $C(z)$ is a low-pass filter or a constant less than 1. It can weaken the interference signal and improve the systematic robustness. The internal mode structure of the odd-order harmonic wave generator is shown in Figure 5.

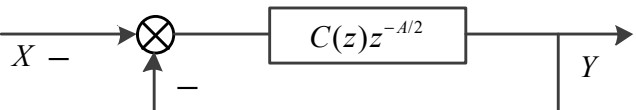

**Figure 5.** Structural diagram of internal model.

The corresponding amplitude–frequency characteristics are shown in Figure 6. Based on the analysis, the internal model can generate amplitude gain at the fundamental wave and each order of harmonic wave, enabling it to track and regulate the harmonic components across the frequency band and effectively filter them out.

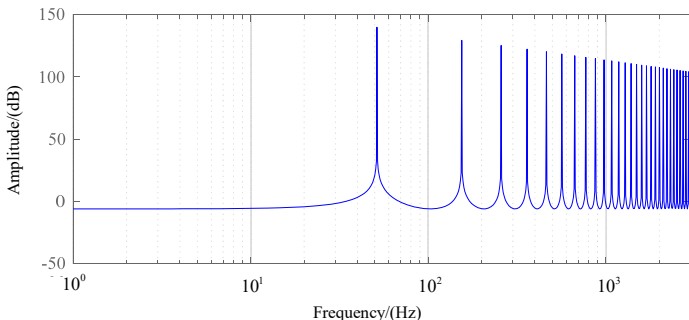

**Figure 6.** The curve of amplitude–frequency characteristics of the internal model.

*3.2. The Strategies for DC Offset Elimination and Frequency Adaptation*

3.2.1. DC Offset Elimination

According to Figure 3, $D(s)$ and $Q(s)$ show different attenuation characteristics concerning the DC component. In the case of DC offset, $u'_{g\alpha}$ and $u'_{g\beta}$ basically contain no DC component. However, due to the low-pass characteristic of the $Q(s)$ filter, $qu'_{g\alpha}$ and $qu'_{g\beta}$ contain a great many DC components. The DC offset is $ku_{gDC}$, which correlates with $k$. In the case of a β-axis voltage, the error signal $\varepsilon$ can be expressed as $\varepsilon = u_{g\beta} - u'_{g\beta} = u_{g\beta DC} + u_{g\beta n}$, in which $u'_{g\beta}$ is the output voltage of the QSG link. The schematic diagram of DC offset elimination is displayed in Figure 7. First of all, the error signal is amplified $k$ times and then processed by a low-pass filter so that $ku_{g\beta DC}$ can be obtained. In this way, the DC component in $qu'_{g\beta}$ is offset, and the orthogonal signal without DC component is obtained. The voltage $u_{g\alpha}$ in the α axis shares the same situation.

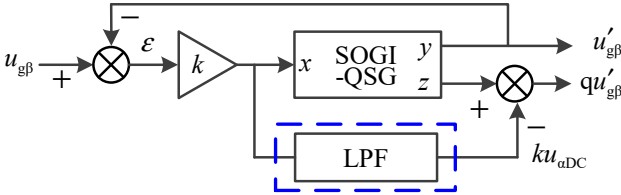

**Figure 7.** Schematic diagram of DC offset elimination.

3.2.2. Frequency Adaptation

In reality, the grid voltage frequency may not always remain at exactly 50 Hz in a power system. The allowable deviation range varies depending on the grade of the power grid. According to [18], the allowable frequency deviation is ±0.2 Hz for a system capacity greater than 300 MW and ±0.5 Hz for a system capacity less than 300 MW. Based on [19], the maximum allowable frequency range is ±1 Hz.

In wind-power systems, it is stipulated that when the grid frequency deviation is within ±3 Hz, the wind-driven generator should remain operating within the grid. Therefore, it is crucial for engineers to ensure that DSOGI-PLL can accurately lock the phase as the grid frequency fluctuates to ensure the stability and reliability of the power system.

As analyzed in Section 3.2, an ideal orthogonal signal will only be output from the SOGI-QSG link when the frequency of the input signal is equal to the set value of $\omega_g$. Otherwise, the phase of the output signal will deviate from that of the input signal. To address this issue, a functional block diagram for frequency adaptation is presented in Figure 8.

In this approach, the frequency $\omega'_g$ detected by DSOGI-PLL is input into SOGI-QSG, and the resonant frequency is regulated in real time so that it conforms to the actual grid frequency. This enables the frequency adaptation to be realized and ensures that ideal orthogonal signals can be accurately obtained even when the grid frequency fluctuates.

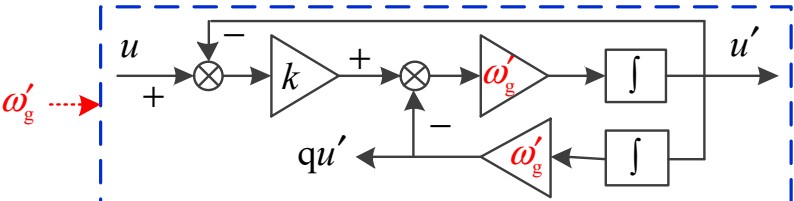

**Figure 8.** Schematic diagram of frequency adaptation.

### 3.2.3. Improved DSOGI-PLL Structure

The improved DSOGI-PLL structure, as shown in Figure 9, is embedded with the external loop of the internal model for repetitive control. To filter out the fundamental wave component, a 50 Hz wave trap is added to the improved structure. This approach enables the harmonic signals across the frequency band to be obtained. These signals are then fed back negatively to the PLL input end to offset the harmonic voltage in the input voltage signal.

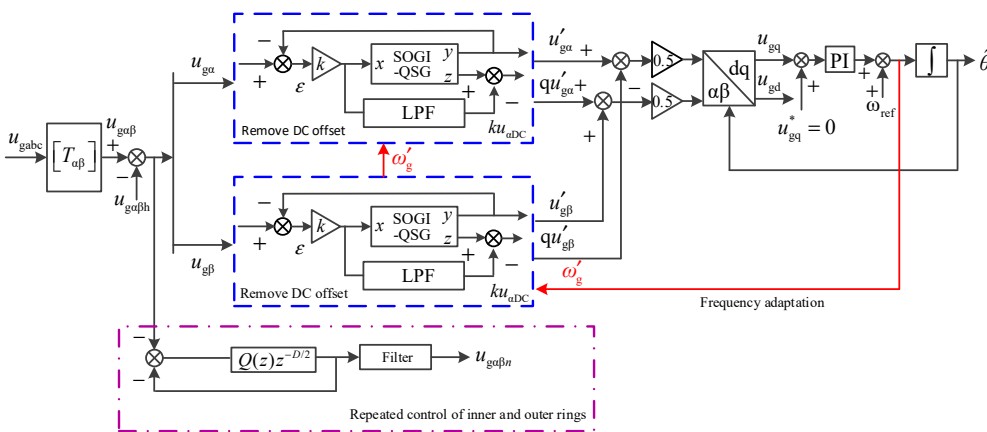

**Figure 9.** Structure diagram of improved DSOGI-PLL.

In addition to harmonic signal filtering, DC offset elimination and frequency adaptive strategies are introduced to ensure accurate phase locking by the improved DSOGI-PLL under conditions of imbalance, harmonic waveform, DC offset, and frequency fluctuations. As such, the improved DSOGI-PLL structure achieves high performance and reliability in practical applications.

## 4. Simulation and Analysis

A system simulation model was built using the Simulink platform to compare and evaluate the performance of the SRF-PLL, traditional DSOGI-PLL, and improved DSOGI-PLL synchronization schemes under nonideal grid conditions. Because the proposed algorithm is an improvement based upon the traditional DSOGI-PLL, DSOGI-PLL was chosen in the simulation comparison. The grid voltage frequency was set to 50 Hz, and the system switching frequency was set to 10 kHz. The system switching frequency is related to the overall design of the system. The higher the switching frequency, the smaller the passive device volume required by the system; however, the corresponding switching loss will also increase. The IGBT module is generally used in the VSC-HVDC system, so the frequency of 10 kHz was selected for overall efficiency.

Through simulation and comparison, the performance of each synchronization scheme can be analyzed and evaluated, including issues such as phase locking accuracy, output waveform distortion, and stability under different operating conditions. The simulation results can provide insights into the strengths and weaknesses of each synchronization scheme and can guide the selection of the appropriate synchronization method for specific power system applications.

*4.1. Comparison between SRF Scheme and Traditional DSOGI Scheme in an Imbalanced Grid*

To compare the phase-locked performance of the SRF-PLL and traditional DSOGI-PLL in an imbalanced grid, a negative-sequence voltage of 30% was injected into the grid at 0.05 s, and the phase angle was simultaneously changed by 20°. The simulation results are shown in Figure 10.

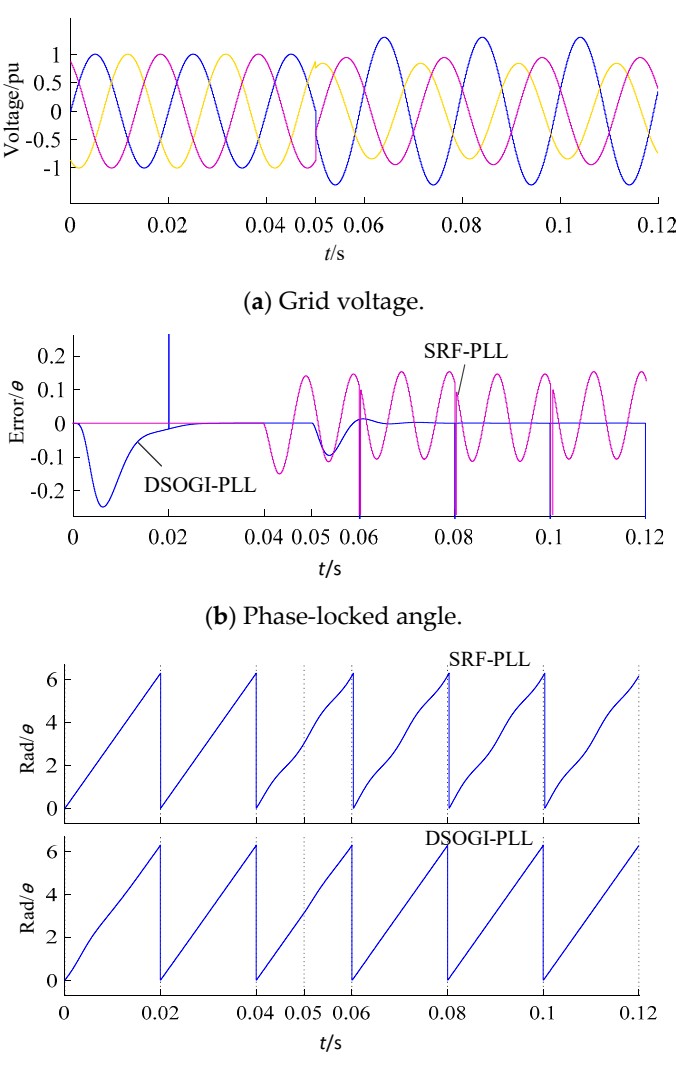

(**a**) Grid voltage.

(**b**) Phase-locked angle.

(**c**) Phase-locked error.

**Figure 10.** Simulation waveform in an imbalanced grid.

Based on the results of the comparison, it can be seen that the phase-locked error of the SRF-PLL is a sinusoidal wave with fluctuations at twice the frequency within the range of ±0.15. On the other hand, DSOGI-PLL can effectively suppress imbalanced voltage interference and its phase-locked error remains at 0, indicating that it can maintain a stable and accurate phase-locking performance even under nonideal grid conditions. Yellow, blue, and purple represent three-phase voltage.

*4.2. Comparison between Traditional DSOGI and Improved DSOGI Schemes in a Nonideal Grid*

4.2.1. Harmonic Power Grid

In another simulation, a 30% quintuple harmonic wave in the negative sequence was injected into the grid at 0.05 s, and the phase angle was simultaneously changed by 50°, as shown in Figure 11a. Figure 11b compares the PLL error waveforms of the traditional DSOGI-PLL and improved DSOGI-PLL in a grid with quintuple harmonics.

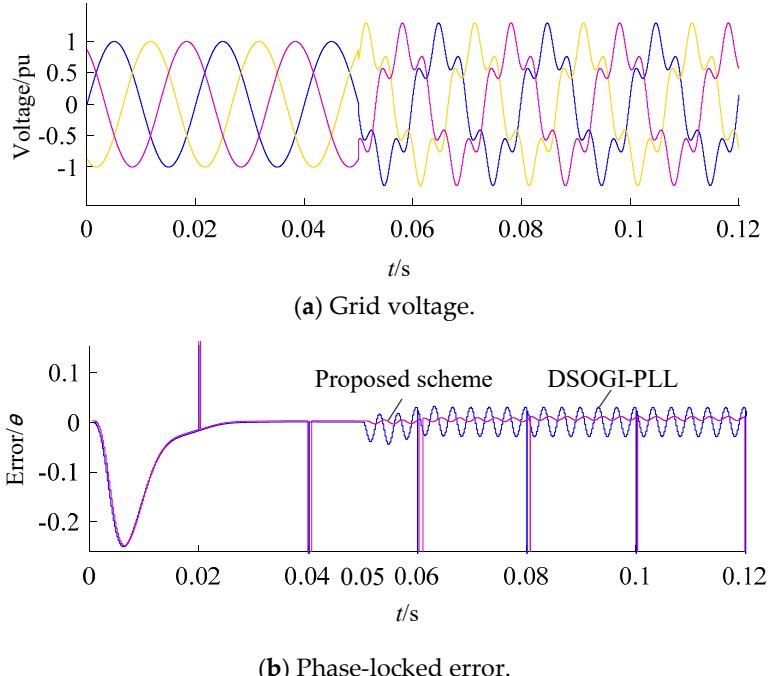

(**a**) Grid voltage.

(**b**) Phase-locked error.

**Figure 11.** Simulation waveform in a harmonic grid.

Although the attenuation characteristic of the SOGI link can reduce harmonic interference to a certain extent, some harmonic pollution cannot be ignored if more accurate phase information is pursued. The simulation results indicate that the phase-locked error of traditional DSOGI-PLL presents small fluctuations at the sixfold frequency with an amplitude of ±0.03, while an improved DSOGI-PLL can completely eliminate the influence from harmonic waves and accurately track the phase information in a harmonic grid without any significant fluctuation.

### 4.2.2. DC Offset Grid

To verify the phase-locked superiority of the improved DSOGI-PLL in a DC bias grid, a 15% DC component was injected into the C-phase voltage of the grid at 0.05 s, as shown in Figure 12a. According to the phase-locked error waveform in Figure 12b, the error signal of the traditional DSOGI-PLL is presented as a sinusoidal wave fluctuating within the range of ±0.08, and its frequency is the fundamental frequency. The captured phase at this moment shows a low-frequency oscillation with an amplitude of about 2.5%. In contrast, in the improved DSOGI-PLL scheme, the phase-locked error occurs upon the injection of the DC component and is no longer 0, but it stabilizes to 0 again after quick regulation. This verifies the highly accurate phase-locked performance of the improved DSOGI-PLL in a DC offset grid. As such, it can maintain phase synchronization with high accuracy even when there are DC components in the grid voltage.

### 4.2.3. Grid Frequency Fluctuation

Although the actual frequency variation in the actual power grid does not allow a wide range of variations, in order to verify the dynamic response of the algorithm for the power grid under extreme conditions, the frequency range was from 50 Hz to 43 Hz. In another simulation, the grid voltage frequency was changed from 50 Hz to 43 Hz at 0.05 s, and the phase-locked accuracies of both PLLs under the grid frequency fluctuation were tested. The comparison of the two output phase-locked error waveforms in Figure 13b indicates that the traditional DSOGI-PLL cannot accurately lock the grid voltage phase information after the frequency changes. Its phase-locked error waveform changes periodically, with a fluctuation amplitude of ±6.28.

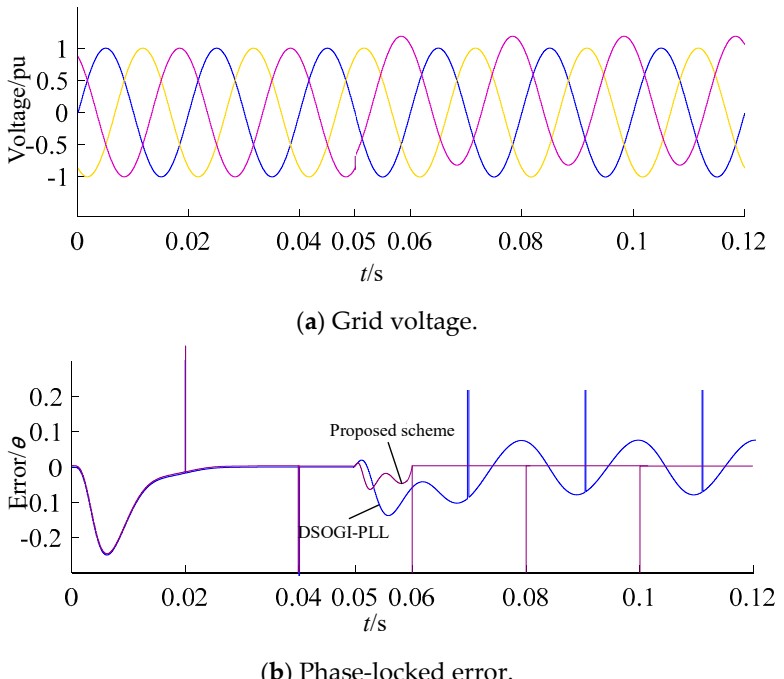

(**a**) Grid voltage.

(**b**) Phase-locked error.

**Figure 12.** Simulation waveform in a DC offset grid.

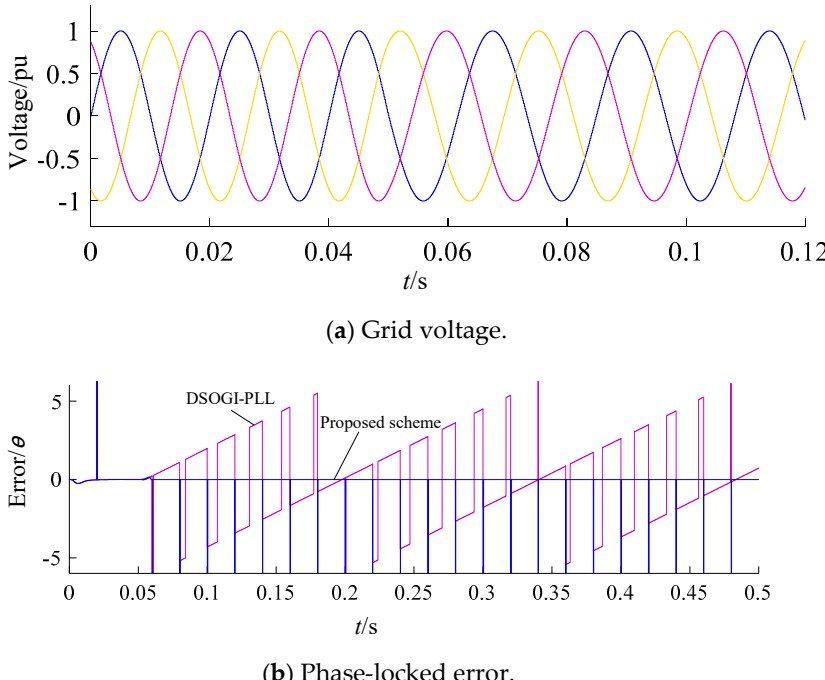

(**a**) Grid voltage.

(**b**) Phase-locked error.

**Figure 13.** Simulation waveform under grid frequency fluctuation.

On the other hand, the improved DSOGI-PLL can still accurately lock the phase information after quick regulation. This demonstrates its superior phase-locked performance even under unexpected frequency fluctuations in practical power systems. As such, the use of the improved DSOGI-PLL can ensure the stability and reliability of power system operation amidst unexpected changes in the grid frequency.

Based on the simulation results under three kinds of nonideal grid conditions, the phase-locked error waveforms of traditional DSOGI-PLL schemes are represented as a sinusoidal wave with fluctuations at the sixfold frequency, a sinusoidal wave with fluctuations at the fundamental frequency, and periodically varying waveforms, respectively.

Figure 14 shows the comparison between the proposed scheme and SRF-PLL and DSOGI-PLL, respectively. The proposed algorithm is characterized by imbalance in the power grid, harmonics in the power grid, DC components in the power grid, and frequency fluctuations. It can be seen that the proposed algorithm has obvious advantages in dealing with frequency fluctuations and DC components.

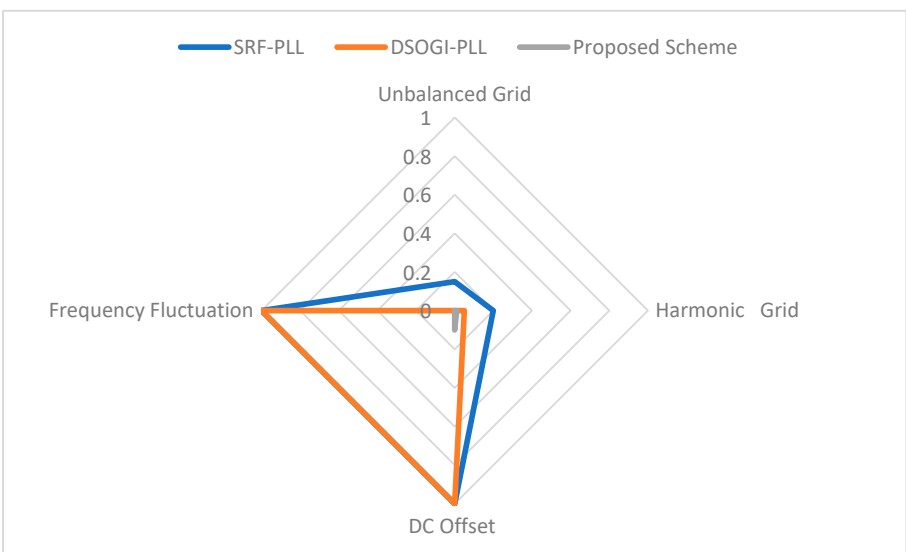

**Figure 14.** Comparison between different algorithms.

## 5. Conclusions

In this study, a VSC-HVDC synchronization scheme featuring an improved DSOGI-PLL was proposed for application to a harmonic grid. The proposed scheme integrated an external loop of an internal model for repetitive control into the original DSOGI structure. Moreover, strategies were introduced to eliminate the DC offset and adapt to frequency fluctuations in the grid. The phase-locked accuracies of the traditional DSOGI-PLL and the improved DSOGI-PLL methods were simulated and compared in several nonideal grid conditions. The simulation results showed that the improved DSOGI-PLL can accurately lock the phase in imbalanced, harmonic, DC offset, and frequency fluctuation grids with high accuracy. The simulation results verified the phase-locked superiority of the improved DSOGI-PLL in such nonideal grids. Additionally, an experimental platform was established to conduct the phase-locking experiment in the harmonic grid. The experimental results further confirmed the effectiveness of the proposed improved DSOGI-PLL synchronization scheme in accurately tracking the grid's phase information, even under nonideal conditions. Overall, the proposed improved DSOGI-PLL synchronization scheme can effectively enhance the accuracy and stability of phase-locking in power systems, mitigating potential safety hazards caused by non-ideal grid conditions while achieving improved power transfer capability. Due to the gradual development of electric power systems in the future, PLL will be further iteratively upgraded for weak resistance power grids in the future. In weakly resistive power networks, the influence of decaying DC cannot be ignored, so the future PLL scheme can further consider the influence of decaying DC.

**Author Contributions:** Conceptualization, L.P.; methodology, Z.F.; software, T.X.; validation, Y.Q.; formal analysis, W.Z.; investigation, C.Z.; writing—original draft preparation, L.P. All authors have read and agreed to the published version of the manuscript.

**Funding:** This research received no funding.

**Data Availability Statement:** Not applicable.

**Conflicts of Interest:** The authors declare no conflict of interest.

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
