# Peer review of "An Improved Dual Second-Order Generalized Integrator Phased-Locked Loop Strategy for an Inverter of Flexible High-Voltage Direct Current Transmission Systems under Nonideal Grid Conditions"

_processes, doi:10.3390/pr11092634_

Round 1

Reviewer 1 Report

Authors presents Improved DSOGI-PLL Strategy for Inverter of Flexible HVDC  Transmission Systems under Non-ideal Grid Conditionintereting research which This paper proposes a PLL adapted to the extremely harsh grid conditions. Firstly, the traditional synchronous reference frame PLL and dual second order generalized integrator (DSOGI-PLL) are analyzed and the errors in phase-locking and the shortcomings of these two methods in the presence of DC components in the grid are pointed out. Secondly, based on the harmonic grid voltage, a repetitive control internal model is introduced by DSOGI to realise the real-time tracking and regulation of the harmonic signals, so as to suppress the harmonic voltage disturbancehowever they must modify in few areas

1. Literature survey must be added after introduction section which readers can know the movelty

2. Novelty of the propsoed work can be imrpoved with existing methods in a comaprative measures

3.Maintain same font in all the figures 

4. Comparative assement can be improved

5. Add system parameters

6. Add latest reasearch articles to support the research

7. Conclusion section can be consied

8. Check for typo errors

Reviewer 2 Report

-Pp.1-2: Introduction: OK;

-Pp.2-6: Section  Traditional DSOGI-PLL Scheme: Put references for eqs.;

- P.5, Fig.3: Are they simulated diagrams? If so, what program was used for the simulation?

- P.7, Fig.6: Are they simulated diagrams? If so, what program was used for the simulation?

- P.8, Row 233: „For example, the allowable frequency deviation is ±0.2 Hz for a system capacity 233 greater than 300 MW, and ±0.5 Hz for a system capacity less than 300 MW.”. Where are these values ​​from? Put the reference.

- P.9, Section Simulations: Why did you choose to do a comparative analysis with DSOGI-PLL schemes?

- P.9, Row 269: Why you choose 10 kHz?

- P.9, Rows 277-278: „To compare the phase-locked performance of the SRF-PLL and traditional DSOGI- 277 PLL in an imbalanced grid, a negative-sequence voltage of 30% was injected into the grid 278 at 0.05 s, and the phase angle was simultaneously changed by 20°”. Are values ​​close to practical reality?

- P.10, Rows 296-298: „In another simulation, a 30% quintuple harmonic wave in the negative sequence was 296 injected into the grid at 0.05 s, and the phase angle was simultaneously changed by 50°, 297 as shown in Fig. 11(a).”. Are values ​​close to practical reality?

- P.12, Rows 330-331: „In another simulation, the grid voltage frequency was changed from 50 Hz to 43 Hz 330 at 0.05 s,…”. Isn't it a bit wide range of frequency change?

- P.13, Section Experiment and Analysis: Present in detail, including pictures, with the experimental model. Improve the quality of this section;

- P.15, section Conclusions: What other researches can be developed starting from the researches carried out?

Minor editing of English language required.

Reviewer 3 Report

Interesting paper  for a improved DSOGI-PLL  strategy for inverter of  flexible HVDC  transmission systems.  I have the following recommendations for the work:

1)       Consider adjusting the layout, such as Figures 10 and 12. Some of the figures (a) are on one page, some on another (b) and are more difficult to match.

2)        In the text when explaining the principle of operation, "high voltage direct current (HVDC) transmission" gets lost in the exposition, focusing on 50 hertz signal processing.

3)       The caption under Figure 3(a) should also contain "Figure 3".

4)       На фигура 14 са показани времедиаграми, предполага се от експеримент, който е описан в точка 4 "Experiment and analysys".  Да се поясни начинът на получаване на диаграмите на фигура 14.

5)       Section 1.2 is entitled "The scheme of .....", is that the most accurate term?

6)       In Figure 10a, clarify the diagram with the yellow and blue colors in the legend at the graph.

7)       To expand the literature review. In the description of the proposed diagram in Figure 9, highlight what is from the literature and what is the authors' contribution.

8)       Figure 13b compares the error of the proposed and existing scheme. Increase the raster along the ordinate to show that the error of the proposed one is not that large.

In conclusion, Interesting research and I propose that it be published.

Round 2

Reviewer 2 Report

- P.8, Row 233: „For example, the allowable frequency deviation is ±0.2 Hz for a system capacity 233 greater than 300 MW, and ±0.5 Hz for a system capacity less than 300 MW.”. Where are these values ​​from? Put the reference.

- P.9, Section Simulations: Why did you choose to do a comparative analysis with DSOGI-PLL schemes?

- P.9, Row 269: Why you choose 10 kHz?

- P.9, Rows 277-278: „To compare the phase-locked performance of the SRF-PLL and traditional DSOGI- 277 PLL in an imbalanced grid, a negative-sequence voltage of 30% was injected into the grid 278 at 0.05 s, and the phase angle was simultaneously changed by 20°”. Are values ​​close to practical reality?

- P.10, Rows 296-298: „In another simulation, a 30% quintuple harmonic wave in the negative sequence was 296 injected into the grid at 0.05 s, and the phase angle was simultaneously changed by 50°, 297 as shown in Fig. 11(a).”. Are values ​​close to practical reality?

- P.12, Rows 296-298: „In another simulation, the grid voltage frequency was changed from 50 Hz to 43 Hz 330 at 0.05 s,…”. Isn't it a bit wide range of frequency change?

- P.13, Section Experiment and Analysis: Present in detail, including pictures, with the experimental model. Improve the quality of this section;

- P.15, section Conclusions: What other researches can be developed starting from the researches carried out?

Minor editing of English language required.
